# Two pathways regulate cortical granule translocation to prevent polyspermy in mouse oocytes

Liam P. Cheeseman[1], Jérôme Boulanger[1], Lisa M. Bond[1] & Melina Schuh[1,2]

An egg must be fertilized by a single sperm only. To prevent polyspermy, the zona pellucida, a structure that surrounds mammalian eggs, becomes impermeable upon fertilization, preventing the entry of further sperm. The structural changes in the zona upon fertilization are driven by the exocytosis of cortical granules. These translocate from the oocyte's centre to the plasma membrane during meiosis. However, very little is known about the mechanism of cortical granule translocation. Here we investigate cortical granule transport and dynamics in live mammalian oocytes by using Rab27a as a marker. We show that two separate mechanisms drive their transport: myosin Va-dependent movement along actin filaments, and an unexpected vesicle hitchhiking mechanism by which cortical granules bind to Rab11a vesicles powered by myosin Vb. Inhibiting cortical granule translocation severely impaired the block to sperm entry, suggesting that translocation defects could contribute to miscarriages that are caused by polyspermy.

[1] Medical Research Council Laboratory of Molecular Biology, Francis Crick Avenue, Cambridge Biomedical Campus, Cambridge CB2 0QH, UK. [2] Max Planck Institute for Biophysical Chemistry, Am Fassberg 11, Göttingen 37077, Germany. Correspondence and requests for materials should be addressed to M.S. (email: melina.schuh@mpibpc.mpg.de).

A viable human embryo can only develop from an egg that is fertilized by a single sperm[1,2]. However, human eggs are frequently fertilized by multiple sperm: around 10% of spontaneous abortions are due to triploidy[3], the presence of an extra set of chromosomes in a fetus, and the majority of triploidy is caused by polyspermy[4,5]. Consistent with these statistics, the rate of polyspermy in *in vitro* fertilized human eggs is around 10% (ref. 1) and therefore remarkably high. Understanding the mechanisms that protect mammalian eggs from polyspermy is thus not only of interest for fundamental research but also of direct medical relevance.

Two primary safeguards protect eggs against polyspermy: firstly, polyspermy is prevented at the plasma membrane level, where fusion of the first sperm causes depolarization of the membrane and shedding of the egg's sperm receptor Folr4 (folate receptor 4; also known as Juno), thereby preventing fusion of further sperm[6,7]. Secondly, fertilization triggers the exocytosis of cortical granules, a process termed 'cortical reaction'. Cortical granules contain enzymes that modify and thereby harden the zona pellucida, a proteinaceous matrix surrounding the oocyte[8]. This lowers the binding affinity of sperm by cleavage of the zona pellucida protein ZP2 (ref. 9) and makes the zona impermeable to additional sperm.

Cortical granules are synthesized in the centre of the oocyte[10] and translocate to the plasma membrane during meiosis in preparation for fertilization. Although the cortical reaction has been observed for many years[11], the mechanisms that underlie the synthesis, transport and release of cortical granules in mammals are still mostly unclear. Research in various species has identified molecular players involved in either the transport[12,13], docking at the plasma membrane[13] or exocytosis[14–18] of the granules, but the majority of players in mammalian oocytes are yet to be discovered.

It has previously been shown that the transport of cortical granules to the plasma membrane is an actin-dependent process[12,13]. Previously, we have demonstrated the existence of a cytoplasmic actin network in mouse oocytes which is able to mediate the long-range transport of Rab11a vesicles[19] and is required for the correct positioning of the meiotic spindle, a process essential for asymmetric division[20]. The actin-based nature of this network, along with its suitability for long-range transport, made it an ideal candidate for the translocation of cortical granules and warranted further investigation.

Here, we show that cortical granules translocate along the cytoplasmic actin network in a process that is regulated by Rab27a. The translocation occurs via two distinct pathways: in the first pathway, myosin Va transports cortical granules along actin. In the second unexpected pathway, cortical granules 'hitchhike' on Rab11a vesicles that move to the plasma membrane, which ultimately delivers the granules to the cortex. We show that the disruption of granule translocation and subsequent association with the plasma membrane in *Rab27a* mutant oocytes causes a dramatic increase in the number of sperm that can penetrate the zona. These results provide the molecular mechanisms underlying cortical granule translocation, and suggest that translocation defects could contribute to the large number of miscarriages caused by polyspermy.

## Results

**Rab27a is a marker for cortical granules in live oocytes.** The expression of GFP-Rab27a in mouse oocytes generated a pattern that was reminiscent of cortical granules. We therefore sought to confirm whether Rab27a colocalized with cortical granules by staining fixed oocytes expressing GFP- or mCherry-Rab27a with rhodamine- or FITC-labelled lens culinaris agglutinin (LCA), an

established marker for cortical granules[21] (Fig. 1a). We found that the vast majority of Rab27a puncta colocalized with LCA (Fig. 1b), suggesting that cortical granules are positive for Rab27a. To verify this, we expressed GFP-Rab27a in live oocytes and imaged them through meiotic maturation. The behaviour of Rab27a spots was strikingly similar to that of cortical granules, including a strong recruitment to the cortex during meiotic maturation (Fig. 1c,f), a reduction of the number of puncta in the centre of the cells (Fig. 1c,e), and the formation of a Rab27a-free zone at the cortex adjacent to the meiotic spindle, similar to the previously observed cortical granule-free zone[21,22] (Supplementary Fig. 1; Supplementary Movie 1). Interestingly, although 25% of Rab27a puncta did not stain positive for LCA, we did not observe a subpopulation that remained in the oocyte centre, suggesting that the LCA-negative granules also displayed the hallmarks of cortical granule behaviour. It is possible that the variability in staining may reflect different stages of granule maturation. We therefore concluded that cortical granules are positive for Rab27a, and that Rab27a is a suitable marker for these granules in live oocytes, consistent with a recent study in fixed cells[16].

Previous studies only analysed the behaviour of cortical granules in fixed oocytes, and therefore could not reveal the dynamics of their behaviour. They also focussed on granules docked at the plasma membrane, with no quantitative information on their number in the oocyte centre. The discovery of a marker for cortical granules in live oocytes now allowed us to address these points. We investigated the dynamics of cortical granules during meiotic maturation, during which they are synthesized and translocated to the plasma membrane. To this end, we recorded 4D videos of oocytes expressing GFP-Rab27a and counted the number of cortical granules in the centre of the cell and measured the intensity of Rab27a at the cell cortex (Fig. 1d). Granules in the centre of the cell produced a clear pattern over time: (1) an initial increase in number until around nuclear envelope breakdown (NEBD); (2) a decrease in number in the centre of the cell by 37% starting around 10 min following NEBD and lasting for around 4.5 h; (3) a second increase in granule numbers in the cytosol, likely due to the generation of new cortical granules in the centre of the cell (Fig. 1e). We observed that the decrease in granule number in the centre of the cell coincided with an increase in Rab27a intensity at the cortex, consistent with the notion of cortical granule translocation to the cell periphery during this time (Fig. 1f). We therefore divided the time of meiotic maturation into three distinct phases based on these observations: phase (1) during which newly synthesized cortical granules accumulated in the cell centre; phase (2) which displays the fastest rate of translocation to the cortex; phase (3) during which cortical granules continue to be recruited to the cortex, but synthesis exceeds translocation. These phases are colour-coded from Fig. 1e onwards. The parameters of cortical granule dynamics are presented in Table 1.

**Rab27a is essential for cortical granule translocation.** We were able to use Rab27a as a marker for cortical granules in live oocytes; however, it remained to be determined whether Rab27a was required for their transport. To investigate this, we fixed and stained oocytes from the Ashen transgenic mouse strain, which lacks a functional *Rab27a* gene[23], with LCA either before NEBD or after being released into meiosis for 5 h, during which time most of the translocation is achieved (Fig. 2a). Remarkably, there was a complete absence of recruitment of cortical granules to the cortex in *Rab27a^{Ash/Ash}* oocytes (Fig. 2a,c). Furthermore, the *Rab27a^{Ash/Ash}* oocytes had significantly more cortical granules in the cell centre before NEBD compared to controls, and this

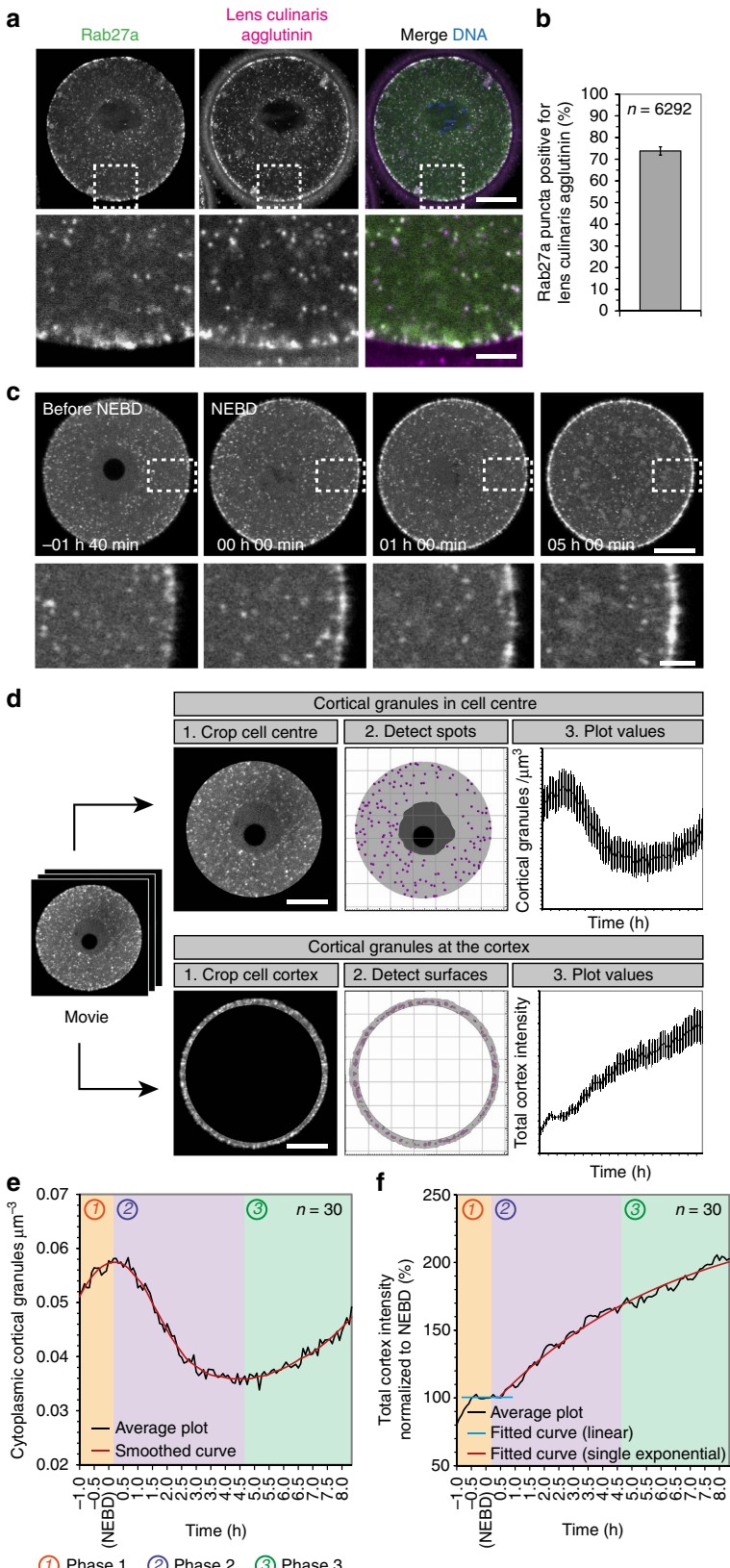

**Figure 1 | Rab27a is a marker for cortical granules. (a)** Oocyte expressing GFP-Rab27a, fixed and stained with lens culinaris agglutinin, a cortical granule marker. Scale bar, 20 μm. **(b)** Quantification of the colocalization of Rab27a puncta with lens culinaris agglutinin puncta (mean ± s.d. from three experiments). Total number of Rab27a puncta quantified are indicated. **(c)** Cortical granules visualized with GFP-Rab27a translocate from the centre of oocytes to the cortex, where they become enriched. Scale bars, 20 μm (overview) and 5 μm (enlarged). **(d)** Schematic diagram of the quantification methods for cortical granule translocation and enrichment at the cortex. Scale bars, 20 μm. **(e)** Cortical granule dynamics in the oocyte centre during meiotic maturation. Three separate phases can be identified in the cell centre, which are colour-coded in all subsequent graphs. The start and end of each phase was determined by calculating the derivative of the data. **(f)** Cortical granule enrichment at the oocyte cortex during meiotic maturation. The three separate phases identified in the cell centre are colour-coded.

**Table 1 | Parameters of cortical granule translocation in mouse oocytes.**

| Parameters in cell centre ($n = 20$ cells) | Value | s.d. | Unit |
|---|---|---|---|
| Onset of phase 2 | 28 | 33 | min |
| End of phase 2 | 246 | 63 | min |
| Duration of phase 2 | 218 | 61 | min |
| Average CG at start of phase 2 per $\mu m^3$ | 0.066 | 0.028 | CGs $\mu m^{-3}$ |
| Average CG at start of phase 2 per oocyte* | 14,573 | 6,176 | CGs |
| Average CG at end of phase 2 per $\mu m^3$ | 0.041 | 0.021 | CGs $\mu m^{-3}$ |
| Average CG at end of phase 2 per oocyte* | 9,092 | 4,648 | CGs |
| Percentage decrease | 37.6 | | % |
| Rate of phase 1 increase per $\mu m^3$ | 0.00016 | 0.00014 | CGs $\mu m^{-3}$ min$^{-1}$ |
| Rate of phase 1 increase per oocyte* | 35.6 | 30.9 | CGs min$^{-1}$ |
| Rate of phase 2 decline per $\mu m^3$ | $-0.00012$ | 0.00012 | CGs $\mu m^{-3}$ min$^{-1}$ |
| Rate of phase 2 decline per oocyte* | $-26.7$ | 27.1 | CGs min$^{-1}$ |
| Fastest rate of decline per $\mu m^3$ (over 60 min period) | $-0.00015$ | 0.00011 | CGs $\mu m^{-3}$ min$^{-1}$ |
| Fastest rate of decline per oocyte* | $-32.1$ | 24.8 | CGs min$^{-1}$ |
| Rate of phase 3 increase per $\mu m^3$ | 0.000056 | 0.000095 | CGs $\mu m^{-3}$ min$^{-1}$ |
| Rate of phase 3 increase per oocyte* | 12.4 | 21.0 | CGs min$^{-1}$ |
| | | | |
| Parameters at the cell cortex ($n = 21$ cells) | | | |
| Onset of increase | 42 | 27 | min |
| Average intensity at start of increase | 99.0 | 16.8 | % |
| Average intensity at end of phase 2 ($n = 14$) | 187.9 | 81.9 | % |
| Percentage increase | 89.8 | | % |
| Average rate of increase of phase 1 | 0.14 | 0.21 | % min$^{-1}$ |
| Average rate of increase of phase 2 | 0.37 | 0.32 | % min$^{-1}$ |
| Average rate of increase of phase 3 | 0.16 | 0.19 | % min$^{-1}$ |
| Rate of change during first 60 min of increase | 0.37 | 0.29 | % min$^{-1}$ |
| Rate of change for 30 min before phase 3 ($n = 14$) | 0.06 | 0.20 | % min$^{-1}$ |
| Fitting equation | $y = A * \exp(-x/t1) + y0$ | | |
| A (Amplitude) | $-123.91$ | 1.90 | |
| t1 (constant decay) | 199.81 | 8.97 | min |
| Upper limit asymptote (y0) | 220.17 | 2.28 | % |
| Adjusted $R^2$ | 0.9897 | | |
| Half-life of exponential increase | 138.50 | | min |

CG, cortical granule.
*Based on oocyte diameter of 75 μm.

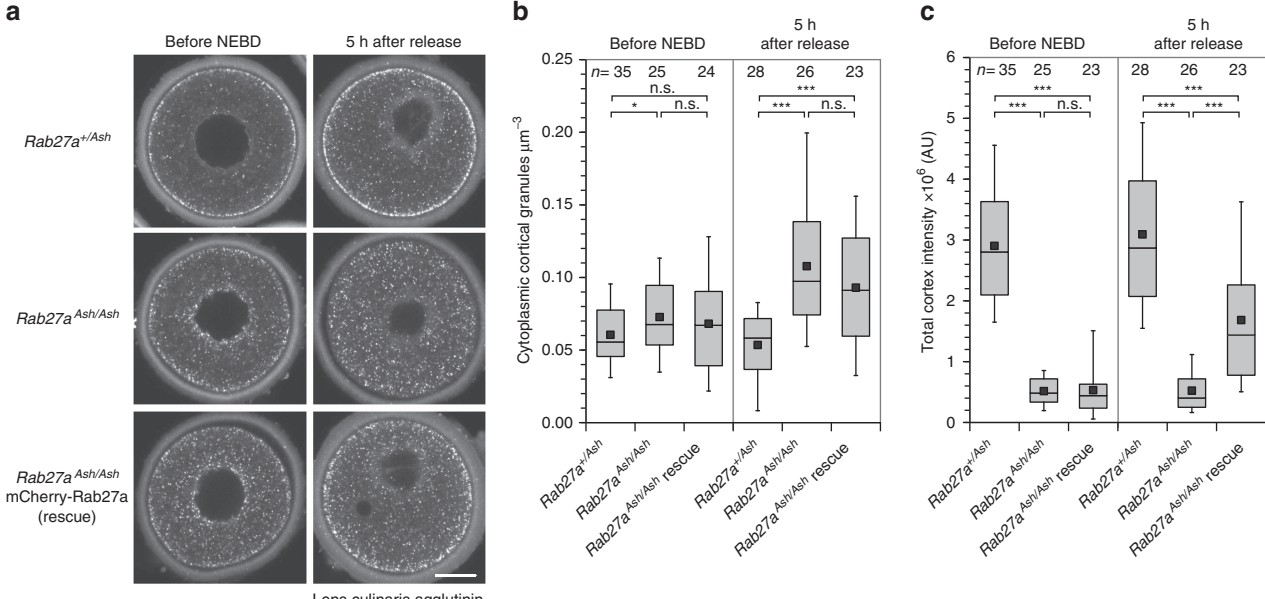

**Figure 2 | Rab27a is required for cortical granule translocation.** (**a**) Maximum intensity projections of confocal micrographs of Rab27a$^{+/Ash}$ (control) or Rab27a$^{Ash/Ash}$ (mutant) oocytes at NEBD or released for 5 h, then fixed and stained with lens culinaris agglutinin. Scale bar, 20 μm. (**b**) Quantification of cortical granules in the cell centre for the conditions in **a**. (**c**) Quantification of cortical granules at the cell cortex for the conditions in **a**. Tukey box plots in **b,c** show the median (line), mean (small square), interquartile range, and the 5th and 95th (whiskers). Significance levels: ns non-significant, *$P < 0.05$, ***$P < 0.001$, analysed using Kruskal–Wallis' ANOVA test from three experiments.

number increased dramatically after a 5-h release, in contrast to control cells (Fig. 2b). Together, these data suggest that Rab27a is involved in the translocation and association of granules with the cortex. We then sought to rescue these defects by expressing functional Rab27a mRNA in the *Ash/Ash* oocytes. This significantly rescued the translocation of cortical granules to the cortex and reduced the number of granules in the cell centre, although these did not reach control levels (Fig. 2a–c). Consistent with our observation that there is little translocation of cortical granules before NEBD, expression of functional Rab27a had no significant effect on oocytes before NEBD. We also observed that in *Ash/Ash* oocytes, cortical granules aggregated in the perinuclear region (see Fig. 2a), which we did not observe in control cells. This indicated a complete lack of transport from their point of synthesis, suggesting that Rab27a was involved in the transport of cortical granules rather than only their maintenance at the cortex. Together, our data indicate that Rab27a is essential for the translocation of cortical granules to the plasma membrane.

**Cortical granule transport via the cytoplasmic actin network.** Previous studies in fixed oocytes indicated that the transport of cortical granules to the plasma membrane is actin-dependent[12,13]. However, detailed studies of the dynamics of cortical granules in the presence and absence of actin in live oocytes were missing. To investigate which stages of the translocation of cortical granules are actin dependent, we treated oocytes with the actin-depolymerizing drug cytochalasin D. In the absence of actin, cortical granule recruitment to the cortex was abolished in all phases of cortical translocation and the number of granules in the cell centre accumulated over time (Fig. 3c,d; Supplementary Fig. 2b; Supplementary Movie 3). By contrast, none of the phases were affected by the microtubule-depolymerizing drug nocodazole, indicating that microtubules are not involved in cortical granule transport (Fig. 3a,b; Supplementary Fig. 2a; Supplementary Movie 2). This suggested that cortical granules are translocated during all phases via an actin-based mechanism with no contribution by microtubules.

Next, we investigated which actin structures and motor proteins mediate the translocation of cortical granules. We and others have previously demonstrated the existence of a cytoplasmic actin network in mouse oocytes that is nucleated by the actin nucleation factors Formin-2 (Fmn2) and Spire1/2 and required for asymmetric spindle positioning in oocytes[24–26]. This network also mediates the transport of Rab11a-positive vesicles over long distances[19]. The movements of these vesicles and the dynamics of the actin network are driven by myosin Vb[19,20]. The actin-based nature of this network, along with its suitability for long-range transport, made it an ideal candidate for the translocation of cortical granules and warranted further investigation. We tested if the translocation of cortical granules relies on the cytoplasmic actin network by imaging cortical granules in oocytes from *Fmn2*$^{-/-}$ mice, which lack the actin nucleator Formin-2 (ref. 26). Indeed, the rate of translocation of cortical granules to the cortex and their reduction in the centre were significantly inhibited (Fig. 3e,f; Supplementary Fig. 2c; Supplementary Movie 4), suggesting that *Fmn2*$^{-/-}$ oocytes are less efficient at translocating cortical granules. We then verified these results by imaging actin and cortical granules together in oocytes expressing GFP-UtrCH and mCherry-Rab27a. Cortical granules were frequently observed moving along actin structures (Fig. 3g) or associated with actin nodes (Fig. 3h). Together, these results show that the cytoplasmic actin network forms the tracks via which cortical granules are transported to the cortex.

**Cortical granules hitchhike on Rab11a vesicles.** We have previously observed that Rab11a vesicles frequently form the nodes of the cytoplasmic actin network[19] and drive the network dynamics[19,20]. As cortical granules sometimes colocalized with the actin nodes (Fig. 3h), we investigated the interactions between Rab11a and Rab27a. Unexpectedly, we frequently noticed a close association between a subset of cortical granules and Rab11a vesicles (Fig. 4a). Upon investigation using high spatial and temporal imaging, we found that a subset of cortical granules seemed to 'latch onto' Rab11a vesicles which passed in proximity to them, then maintained this bond during the rapid Rab11a vesicle movement, and detached several microns from their point of origin (Fig. 4b). These interactions were transient, but could last for more than 10 s (Fig. 4c; Supplementary Movie 7). Additionally, a single Rab11a vesicle was able to bind multiple cortical granules at a time (Fig. 4c). We observed around 7% of cortical granules were bound to Rab11a vesicles at any one time, and there were on average two 'on' and two 'off' events per Rab11a vesicle per minute (Fig. 4d).

We reasoned that this co-transport mechanism could potentiate the translocation of cortical granules since Rab11a vesicles move towards the cortex[19]. We therefore analysed the movements of Rab27a vesicles associated with or dissociated from Rab11a vesicles. Both the speed and the displacement speed of cortical granules were significantly increased when associated with Rab11a, and were of intermediate value between those of Rab27a alone and of Rab11a vesicles themselves (Fig. 4e,f). We also found that 'hitchhiking' Rab27a vesicles travelled further than the 'non-hitchhiking' ones during the same time period (Fig. 4g). Taken together, these data suggest a surprising function for Rab11a vesicles: they transiently bind cortical granules and increase their translocation speed by pulling them towards the cortex.

If Rab11a hitchhiking has a significant impact on cortical granule translocation, one would expect that blocking Rab11a function would decrease the efficiency of translocation. To test this, we expressed the dominant-negative variant Rab11a$^{S25N}$. Cells expressing mutant Rab11a did not display the typical phase 2 decrease in granule number, but interestingly did not accumulate them in the cell centre either: rather the number reached a plateau around NEBD (Fig. 4h,i; Supplementary Fig. 2d; Supplementary Movie 5). Conversely, some cortical enrichment of Rab27a was observed, but this was significantly lower than in wild-type cells. Together, these results suggest that cortical granules are able to accelerate their transport to the cortex by hitchhiking on Rab11a vesicles.

**Anomalous diffusion of cortical granules by actin dynamics.** When Rab27a particles were not hitchhiking on Rab11a vesicles, we often observed them undergoing local movement in random directions similar to diffusion; however, their speed was high and inconsistent with Brownian motion. We wondered if the increased velocity of this anomalous super-diffusion could be due to the association of Rab27a vesicles with the actin network. To test this possibility, we treated oocytes with cytochalasin D to disassemble the actin cytoskeleton and compared the movements of the granules to diffusion-only granule trajectories in control oocytes. The mean speed and displacement speed of cortical granules in oocytes treated with cytochalasin D were significantly lower than those of the diffusive motions observed in controls (Supplementary Fig. 3a–d; $P < 0.0001$ in both cases, Student's *t* test). Local mean squared displacement (MSD) analysis showed that the median diffusion parameter in control oocytes was four times that of cytochalasin D-treated oocytes (Supplementary Fig. 3e,f; 0.0129 versus 0.0031, respectively). This indicated that

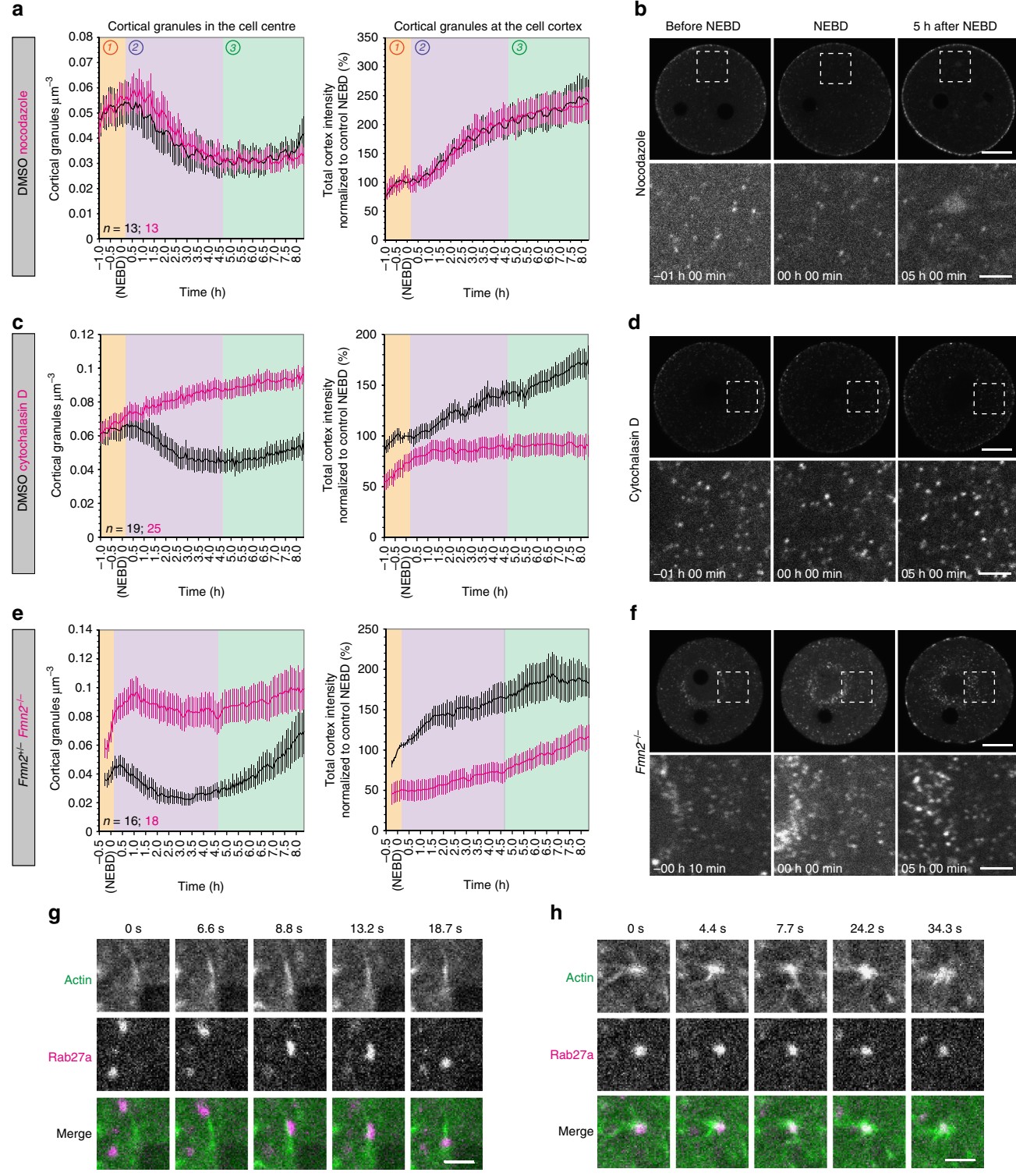

**Figure 3 | Cortical granule translocation is dependent on the cytoplasmic actin network.** Quantification of cortical granule translocation in the oocyte centre or cortex (mean ± s.e.m.) in cells treated with nocodazole or DMSO (**a**), cytochalasin D or DMSO (**c**), in $Fmn2^{-/-}$ (which lack an important actin nucleator) or $Fmn2^{+/-}$ oocytes (**e**). Representative still images (all maximum intensity projections of confocal Z sections) for each condition are presented in **b**,**d** and **f**. Control images are included in Supplementary Fig. 2. Scale bars, 20 μm (overview) and 5 μm (enlarged). (**g-h**) Still confocal images of a cortical granule being transported along actin (**g**), or interacting with an actin node (**h**). Scale bars, 2 μm.

the anomalous diffusion of granules is a complex motion composed of the superimposition of Brownian motion and actin-generated movement.

To investigate further if the high dynamics of cortical granules could be due to association with the dynamic actin network, we blocked the network dynamics by expressing dominant-negative myosin Vb[19]. Myosin Vb is essential for the actin network dynamics, and its inhibition produces a static cytoskeleton[19,20]. The diffusion parameter of non-directed motion was significantly reduced in cells expressing dominant-negative myosin Vb and

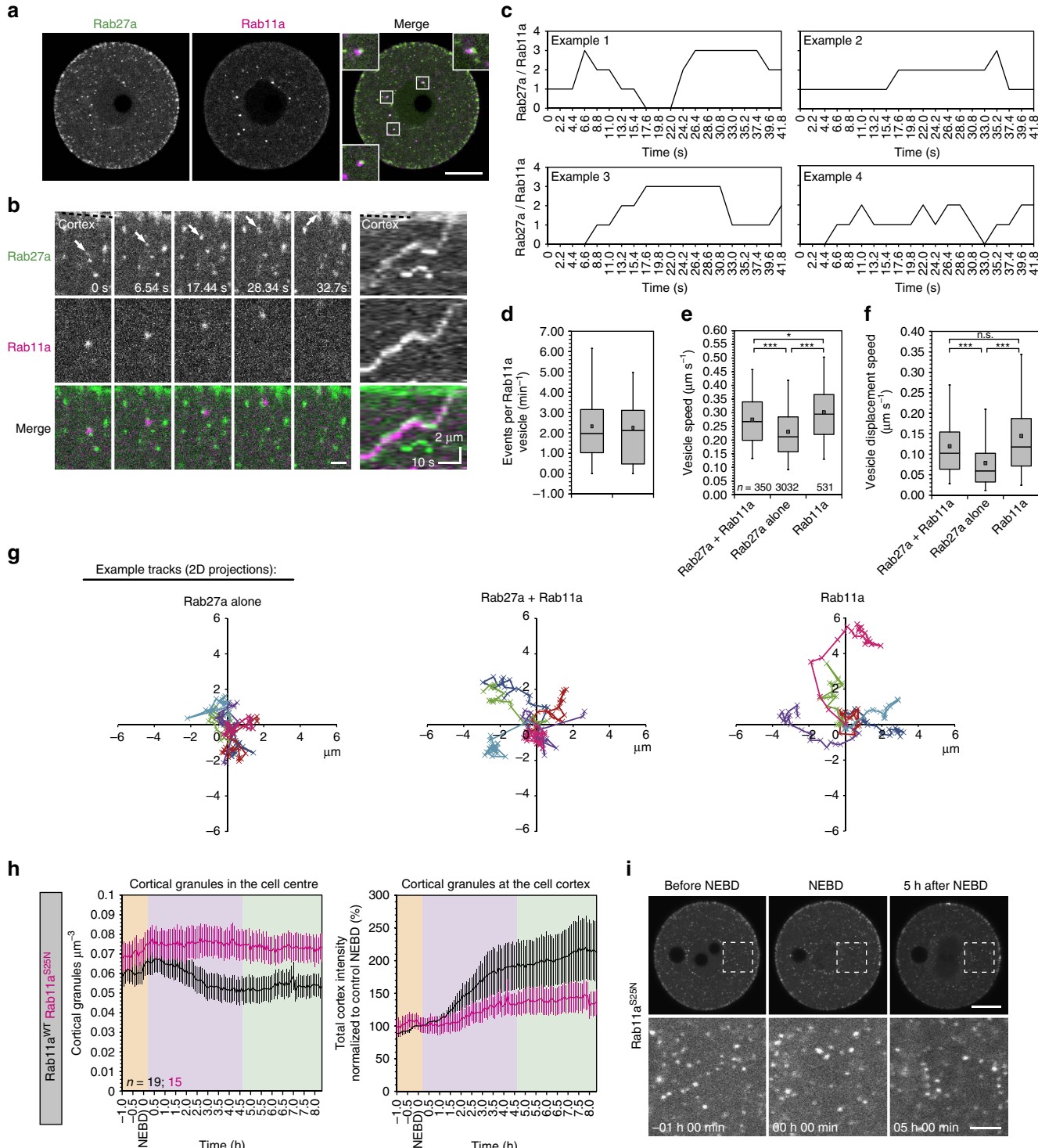

**Figure 4 | Cortical granules are translocated to the cortex by hitchhiking on Rab11 vesicles.** (**a**) Maximum intensity projections of confocal micrographs of an oocyte expressing GFP-Rab27a and mCherry-Rab11a. Note the close association between both types of vesicles (insets). Scale bar, 20 μm. (**b**) Example still images of a cortical granule (Rab27a, green; indicated by arrows) hitchhiking on a Rab11a vesicle (magenta) to reach the cortex (dotted line), with corresponding kymograph. Scale bar, 2 μm. (**c**) Quantification of the number of cortical granules associated to four example Rab11a vesicles. (**d**) Number of cortical granule hitchhiking 'on' and 'off' events per Rab11a vesicle per minute (n = 23 trajectories from three oocytes). Vesicle speed (**e**) and vesicle displacement speed (**f**) were quantified for cortical granules while associated to Rab11a vesicles (Rab27a + Rab11a), cortical granules while not associated to Rab11a vesicles (Rab27a alone), and for Rab11a vesicles. Number of tracks used for quantification are indicated in **e**, from six oocytes. Tukey box plots in **d**,**e** and **f** show the median (line), mean (small square), interquartile range, and the 5th and 95th (whiskers). Significance levels: *$P < 0.05$, ***$P < 0.001$, analysed using Kruskal–Wallis' ANOVA test from two experiments. (**g**) Representative example trajectories of the particle categories indicated in **e** and **f** (six trajectories per condition), projected in two dimensions. Duration of each trajectory is 52 s. (**h**) Quantification of cortical granule translocation in the oocyte centre or cortex (mean ± s.e.m.) in oocytes expressing Rab11a^S25N or wild-type Rab11a. (**i**) Representative still images (maximum intensity projections of confocal Z sections) of oocytes expressing Rab11a^S25N. Control images are included in Supplementary Fig. 2. Scale bars, 20 μm (overview) and 5 μm (enlarged).

anomalous diffusion was largely inhibited (Supplementary Fig. 3g,h), demonstrating that this motion was driven by actin dynamics.

Consistently, cortical granules could associate with actin filaments (Supplementary Fig. 3i) or actin nodes (Fig. 3h) without directed motion, instead displaying diffusive motion. This movement was visibly different to active transport along an actin filament (Fig. 3g). These results suggest that cortical granules are predominantly bound to the actin cytoskeleton between periods of directed transport, consistent with a previous study that showed that the granules link to actin at the onset of meiotic maturation[12].

**Myosin Va transports cortical granules.** Rab27a is known to regulate the transport of secretory cargoes in various cell types in complex with the motor protein myosin Va[27–29]. We investigated whether myosin Va might similarly be involved in the transport of cortical granules by expressing the dominant-negative tail construct MyoVa$^{LT}$, which did not affect Rab11a motility in a previous study[20]. We note that the MyoVa$^{LT}$ construct used here is reported to bind dynein light chain 8 (refs 30,31). However, given that our results demonstrated clearly that the translocation of cortical granules is independent of microtubules (Fig. 3a,b), this potential secondary effect of MyoVa$^{LT}$ is unlikely to have an influence on cortical granule behaviour. Cells expressing MyoVa$^{LT}$ displayed a dramatic accumulation of cortical granules in the cell centre compared to control cells and cells expressing the wild-type myosin (Fig. 5a,b; Supplementary Fig. 2e; Supplementary Movie 6). Intriguingly, MyoVa$^{LT}$ first caused a clustering of cortical granules at the plasma membrane. Shortly after NEBD, the granules were released from the cortex and displayed retrograde movement towards the cell centre (Supplementary Fig. 4). Consistently, the intensity of Rab27a at the cortex declined over time (Fig. 5a,b). Together, these results suggested that myosin Va is involved in the association of cortical granules with the plasma membrane, and potentially in their transport.

We further noticed that in high temporal resolution movies of oocytes expressing GFP-Rab27a and mCherry-Rab11a, cortical granules were able to undergo fast and brief bursts of directed motion towards the cortex in the absence of Rab11a association (Fig. 5c; Supplementary Movie 8). These fast motions still took place when myosin Vb was inhibited, suggesting that a different motor powered their movement separately from myosin Vb and Rab11a (Supplementary Movie 9). We therefore queried whether this myosin Vb- and Rab11a-independent pathway might be powered by myosin Va.

To test if both myosin Va and myosin Vb were involved in cortical granule translocation, we depleted each protein using siRNA in oocytes cultured from follicle cells. The cells were then fixed either at GV stage, or after being released from prophase arrest for 5 h, and then stained with LCA (Supplementary Fig. 6a). We found that depletion of either myosin caused the number of remaining cortical granules in the cell centre to increase significantly compared to controls (Supplementary Fig. 6b). Also, the intensity at the cortex was also lower than in controls when either myosin was depleted (Supplementary Fig. 6c), although significance was not attained due to high staining variability between cells. These results suggested both myosin Va and myosin Vb are involved in the translocation of cortical granules.

Myosin Va-dependent motions have been described as very short bursts of fast movement, and are thus very difficult to discern in living cells[32]. We were unable to identify these motions and to measure their instantaneous speed reliably in the presence

of the high anomalous diffusion of Rab27a granules in oocytes (Supplementary Fig. 3). We therefore used MSD to analyse the velocity parameter separately from the diffusion parameter. Particle trajectories, separated based on whether they displayed Rab11a-dependent or Rab11a-independent active motions, were analysed by local MSD (Fig. 5d; Supplementary Movie 10). As periods of directed motion were extremely transient ($<10$ s), the time lag was set to 6 timepoints (12.6 s) to detect local variations in movement and to dissect the different motions of a trajectory temporally. Velocity and diffusion parameter thresholds were used to separate trajectory steps into types of motions so that we could focus our analysis on periods of directed motion alone (Supplementary Fig. 5; see Methods for details).

When MyoVa$^{LT}$ was expressed, the proportion of active steps in Rab11a-independent trajectories was reduced by over 40% (Fig. 5e,f; $P < 0.0001$, Fischer's exact test). Additionally, the median velocity parameter of Rab11a-independent transport was significantly reduced, suggesting a function of myosin Va in Rab11a-independent transport (Fig. 5h; $P < 0.001$, Mann-Whitney's $U$-test). By contrast, MyoVa$^{LT}$ did not affect the proportion of active steps or velocity parameter of Rab11a-dependent trajectories (Fig. 5e,g; $P = 0.803$, Fischer's exact test; Fig. 5i; $P = 0.376$, Mann-Whitney's $U$-test). As a control, we inhibited myosin Vb which is known to power the movement of Rab11a vesicles[19]. Expression of MyoVb$^{LT}$ strongly affected all types of motion, reducing the number of active steps and both MSD parameters (Fig. 5e–i; Supplementary Fig. 5), consistent with its role in driving the dynamics of the actin network[19,20]. Importantly, it was the only condition to significantly affect Rab11a-dependent transport (Fig. 5g,i), indicating that Rab11a hitchhiking is powered by myosin Vb but not myosin Va, consistent with previous findings[19,20].

An alternative interpretation is that the hitchhiking is merely a by-product of random encounters between Rab11a and Rab27a. However, the inhibition of Rab11a or myosin Vb both prevented efficient granule translocation (Fig. 4h,i; Supplementary Fig. 6), indicating that Rab11a hitchhiking actively contributes to the transport of cortical granules and is a bona fide pathway.

Collectively, these results establish the existence of two separate pathways to translocate cortical granules to the oocyte cortex. The first is a myosin Va-dependent transport mechanism along actin filaments, and the second is an unexpected vesicle hitchhiking mechanism by which cortical granules bind to Rab11a vesicles powered by myosin Vb.

**Disrupting granule transport inhibits the zona pellucida block.** With these results at hand, we were able to investigate the consequences of perturbing the translocation of cortical granules on fertilization. For this, we performed *in vitro* fertilization assays using oocytes from the *Ashen* transgenic mouse line which lacks a functional Rab27a allele, and cannot translocate cortical granules or maintain them at the cortex (Fig. 2). We noticed that fertilized *Rab27a*$^{Ash/Ash}$ oocytes had extra sperm heads that had successfully crossed the zona pellucida (Fig. 6a). This indicated that these cells had an inefficient zona block following fertilization. Upon quantification, almost 90% of all fertilized mutant oocytes had extra sperm cells inside the zona pellucida, compared to less than 20% of heterozygote controls (Fig. 6b; $P < 0.0001$, Student's *t*-test). Moreover, the majority of fertilized *Rab27a*$^{Ash/Ash}$ eggs had five or more extra sperm heads in the perivitelline space, indicating that the defect was severe (Fig. 6c). Subsequent staining with LCA also revealed that the mutant oocytes had a two-fold increase in the number of cortical granules remaining in the cell centre, indicating that the cortical reaction must have been incomplete (Fig. 6d). These results indicate that Rab27a, without

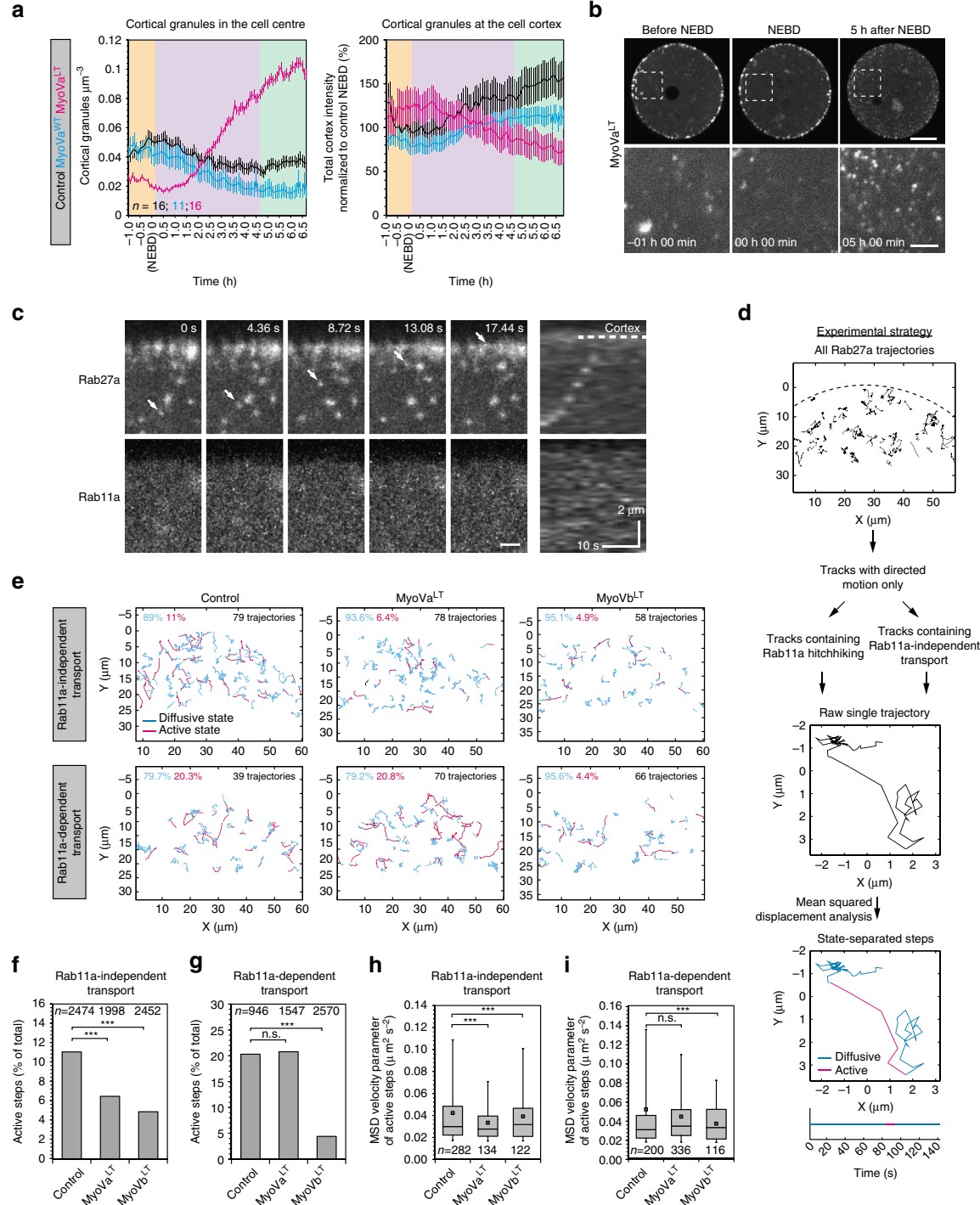

**Figure 5 | A myosin Va-dependent pathway powers cortical granule translocation.** Quantification of cortical granule translocation in the oocyte centre or cortex (mean ± s.e.m) in cells expressing either wild-type myosin Va (MyoVa$^{WT}$), a dominant-negative mutant (MyoVa$^{LT}$), or control (**a**), with representative example still images (maximum intensity projections of confocal Z sections) in **b**. Still images from MyoVa$^{WT}$-expressing cells are included in Supplementary Fig. 2. Scale bars, 20 µm (overview) and 5 µm (enlarged). (**c**) Example still images of a cortical granule (Rab27a) translocating to the cortex in a Rab11a-independent manner (indicated by arrows), with corresponding kymograph. Scale bar, 2 µm. (**d**) Schematic diagram of the local mean squared displacement analysis of Rab27a trajectories that contained Rab11a-dependent transport (hitchhiking) or Rab11a-independent transport, and excluding purely diffusive trajectories. Each track was analysed by MSD with a short lag time to distinguish between active (magenta) and diffusive (blue) states on a step-by-step basis. (**e**) Combined Rab27a trajectories that contained Rab11a-dependent transport (hitchhiking) or Rab11a-independent transport in oocytes expressing MyoVa$^{LT}$, MyoVb$^{LT}$, or control, after local MSD analysis. Active states are coloured in magenta, while diffusive states are in blue. Proportion of each state is indicated (%). (**f–g**) Total number of active steps from Rab27a trajectories (as determined by MSD analysis; % of total steps) containing either Rab11a-independent transport (**f**), or Rab11a-dependent transport (hitchhiking; **g**) in oocytes expressing MyoVa$^{LT}$, MyoVb$^{LT}$, or control. Total number of steps in each category is indicated. Compared using Fischer's exact test. (**h–i**) Mean squared displacement velocity parameters of the active steps from Rab27a trajectories containing Rab11a-independent (**h**) or Rab11a-dependent transport (**i**) in oocytes expressing MyoVa$^{LT}$, MyoVb$^{LT}$, or control. Number of active steps in each category is indicated. Tukey box plots in **h** and **i** show the median (line), mean (small square), interquartile range, and the 5th and 95th (whiskers), compared using Mann-Whitney's U test. Significance levels: ns, non-significant, ***P < 0.001, from three experiments.

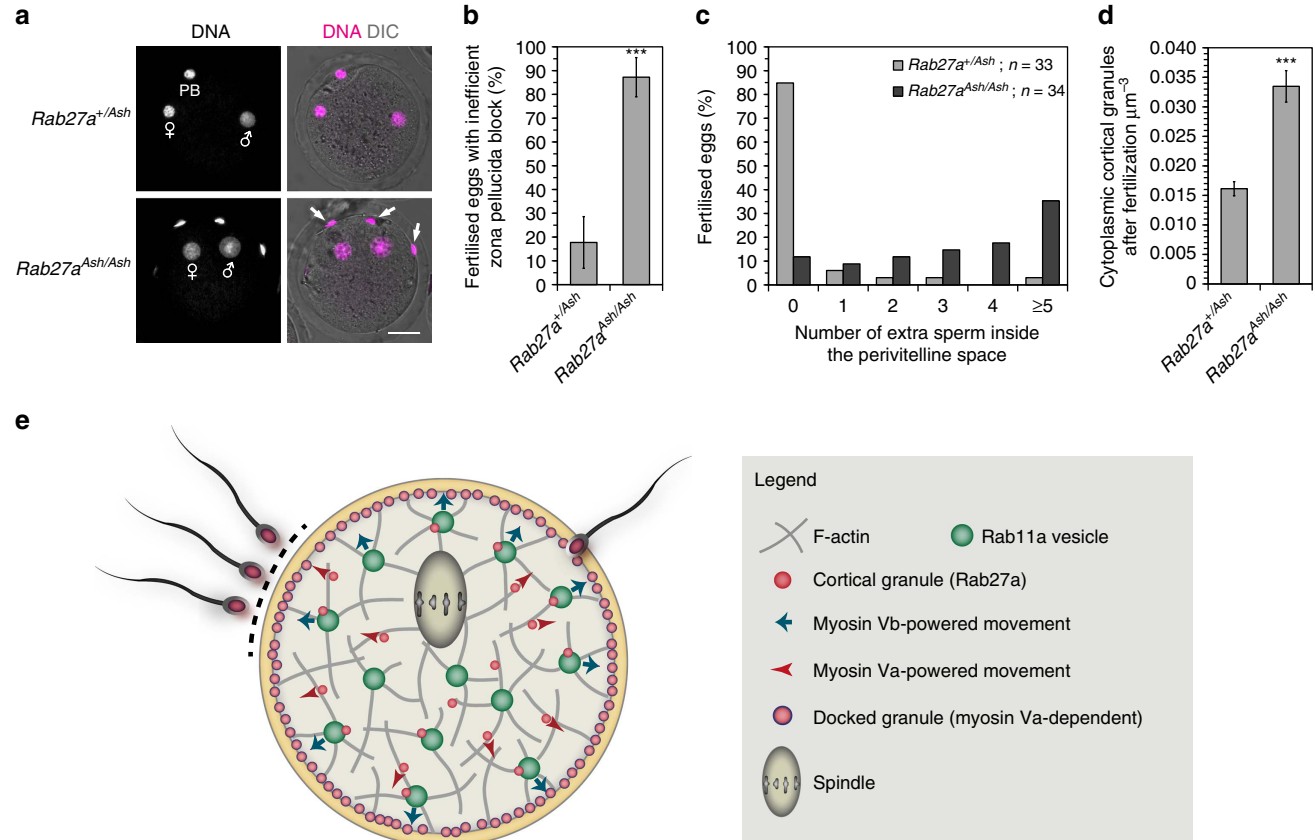

**Figure 6 | Disruption of cortical granule translocation severely impairs the zona pellucida block to polyspermy.** (**a**) Maximum intensity Z projections of *in vitro* fertilized oocytes from *Rab27a$^{+/Ash}$* (control) or *Rab27a$^{Ash/Ash}$* (mutant) oocytes, fixed and stained with Hoechst. Male and female pronuclei and polar body (PB) are indicated. Note the extra sperm heads present inside the zona pellucida in mutant oocytes (arrows). Scale bar, 20 μm. (**b**) Quantification of the number of fertilized oocytes which had extra sperm heads in the perivitelline space in *Rab27a$^{+/Ash}$* or *Rab27a$^{Ash/Ash}$* oocytes (mean ± s.d.; Student's *t* test) (**c**) Quantification of the number of extra sperm inside the perivitelline space of fertilized *Rab27a$^{+/Ash}$* or *Rab27a$^{Ash/Ash}$* oocytes. (**d**) Quantification of remaining cortical granules in the centre of Rab27a$^{+/Ash}$ or Rab27a$^{Ash/Ash}$ oocytes after fertilization (mean ± s.d.; Student's *t* test). (**e**) Mechanistic model of the two cortical granule translocation mechanisms in mammalian oocytes. See text for details. Significance levels: ***$P < 0.001$, from three experiments.

which translocation and maintenance of cortical granules at the cortex are impaired (Fig. 2), has a strong impact on the efficiency of the zona block following fertilization.

## Discussion

In summary, this study provides the first dynamic data on cortical granule translocation in mouse oocytes, and answers the long-standing question of its underlying mechanisms in the form of two pathways (Fig. 6e).

Unexpectedly, we found that cortical granules hitchhike on Rab11a vesicles powered by myosin Vb to reach the cortex. Recent studies reported that large ribonucleoprotein complexes (mRNPs), peroxisomes, lipid droplets and the endoplasmic reticulum are similarly able to hitchhike on early endosomes in some fungal species[33–35] in order to achieve long-range transport. This study provides the first observation of vesicle hitchhiking in mammalian cells, indicating that such transport systems are evolutionarily conserved, and more widely involved in important biological processes than previously anticipated.

In the second mechanism, cortical granules undergo short bursts of movements that are myosin Va-dependent. Both myosin Va (refs 36–41) and Rab27a (refs 42–46) have been implicated in secretory granule trafficking and exocytosis in a number of cells types. In several cases, both proteins form a complex together to

transport cargo[27,28,42,47–53]. This study reinforces a general role of the cooperation between these proteins in organelle trafficking and regulated exocytosis. Since Rab27a and myosin Va are not thought to bind directly[27,43,49,50], it is likely that an adaptor links the two proteins, similarly to the tripartite complexes reported in melanocytes and cytotoxic T lymphocytes[27,45,48–50,52,54]. The identification of the adaptor protein will be of importance in future studies, and raises the interesting possibility that a mouse mutant for this adaptor would display severe defects in the efficiency of the zona block to sperm entry.

The short periods of active transport were interrupted by long periods in which the cortical granules underwent anomalous diffusion. Our results suggest that these unusual dynamics are driven by the highly dynamic actin network that the cortical granules are associated with. This activity likely reflects a highly dynamic actin network. The dynamics of the actin network could potentiate granule translocation by increasing the likelihood of interaction with Rab11a vesicles and of capture by the actin cytoskeleton via a search-and-capture mechanism. Previously, we have suggested a model whereby filaments of the actin network move outward towards the cortex over time, pulled by Rab11a vesicles[19]. If correct, the association of cortical granules to actin would produce a gradual translocation to the periphery. Given the large size of oocytes and the short time window during which

they must undergo meiotic maturation, such a slow but overarching mechanism to boost delivery speed would be highly advantageous.

A previous study reported a 20–50% reduction in litter size in *Ashen* female mice compared to their background strain[55]. Our results now offer an explanation for this observation. It is tempting to speculate that separate redundant polyspermy avoidance mechanisms, such as the block at the plasma membrane[6], are responsible for the remaining fertility of the *Ashen* mice and without which polyspermy would be far more prevalent. The *Ashen* genotype possesses a human equivalent in the form of the autosomal Griscelli syndrome type 2 (refs 56,57). Type 2 Griscelli syndrome patients exhibit partial albinism and immunodeficiency; however, there have been no systematic studies of fertility in these patients.

Our results now provide a conceptual framework for how cortical granules become arranged before fertilization and provide a basis for studying the mechanisms that prevent polyspermy in humans.

## Methods

**Preparation and culture of oocytes.** All mice were maintained in a specific pathogen-free environment according to UK Home Office regulations. The use of animals was approved by the MRC Cambridge Ethical Review Committee, and was performed under the UK Home Office project license 70/8087. Oocytes were isolated from ovaries of 8- to 12-week-old FVB/n, $Fmn2^{-/-}$ or $Rab27a^{Ash/Ash}$ mice, cultured and microinjected as described in detail previously[26]. Briefly, oocytes were collected from the ovaries of 7–14-week-old females by puncturing using a hypodermic needle, and kept in M2 medium containing dibutyryl cAMP (Sigma-Aldrich) at 37 °C. Oocytes were released into meiosis by rinsing into dibutyryl cAMP-free medium. In some experiments, oocytes were treated with 1 μM nocodazole or 5 μg ml$^{-1}$ cytochalasin D (Calbiochem). For *in vitro* fertilization experiments, we followed a previously described protocol[58]. Briefly, mature oocytes from super-ovulated female mice were stripped of cumulus cells by 5-min incubation in M2 medium containing hyaluronidase (0.3 mg ml$^{-1}$). The eggs were then incubated in human tubal fluid medium (Gibco) for 5 h at 37 °C with fresh sperm ($1 \times 10^6$ motile sperm per ml) which had been capacitated for 30 min at 37 °C.

**Follicle culture and siRNA knock down.** To deplete myosin Va or myosin Vb, siRNA knockdown was performed in cultured follicle cells as described previously[59]. Briefly, mixes of three siRNA oligos per target (Qiagen) were microinjected into follicles from 10- to 12-day-old (C57BL × CBA) F1 female mice, and cultured for 8–10 days. siRNA sequences used were (1) ATGATAAATACT GTTATTAAA, (2) CACGATTGTTATTCAGTCTTA and (3) CAGCCTTGTAT CAATCTTATA for myosin Va, and (1) CCGGAAGGTGGATTTGTTAAA, (2) TCAAACTGAGATAATATTAAA and (3) AACCTGGAGTTTCTCAATGAA for myosin Vb. Resulting mature oocytes were then stripped, injected with reporter mRNA and left to express for 3–4 h before release from prophase arrest.

**Confocal microscopy.** Images were acquired with a Zeiss LSM710 confocal microscope equipped with a Zeiss environmental incubator box or a Zeiss LSM780 confocal microscope equipped with a Tokai Hit Stage Top Incubator, with a × 40 C-Apochromat 1.2 NA water-immersion objective lens. Typically, whole cells were imaged by taking a Z-stack of three 0.65 μm-thick sections every 5 or 10 min for 12 h. For high spatial and temporal imaging, Z-stacks composed of three or six 0.65 μm-thick sections were acquired every ~2–2.5 s for 5 min. Images of actin and Rab27a were imaged every 1.1 s in single planes. All images of oocytes shown in this study are maximum intensity projections of the acquired Z sections. Oocytes expressing SNAP-Rab11a were incubated in 647-SiR dye (New England Biolabs) according to the manufacturer's instructions prior to imaging.

**Immunofluorescence microscopy.** Oocytes were fixed for 30–60 min at 37 °C in 100 mM HEPES (pH 7; titrated with KOH), 50 mM EGTA (pH 7; titrated with KOH), 10 mM MgSO$_4$, 2% formaldehyde and 0.2% Triton X-100, based on previously published methods. Oocytes were left in PBS with 0.1% Triton X-100 overnight at 4 °C. LCA staining was carried out in PBS with 0.1% Triton X-100 and 3% BSA. DNA was stained with 5 mg ml$^{-1}$ Hoechst33342 (MolecularProbes).

**Expression constructs and mRNA synthesis.** pEGFP–Rab27a was a gift from M. Seabra and described in Hume *et al.*[43] Rab27a was inserted into a pGEMHE–EGFP or pGEMHE–mCherry vector via XhoI and BamHI. pGEMHE–SNAP–Rab11a$^{S25N}$ was obtained from pGEMHE–EGFP–Rab11a$^{S25N}$ by switching EGFP with SNAP from pSNAPf (New England Biolabs) via

HindIII-XhoI. Full-length pGEMHE–mCherry–myosin-Va, pGEMHE–mCherry–myosin-Va$^{LT}$, pGEMHE–mCherry–myosin-Vb$^{LT}$, pGEMHE–mCherry–Rab11a, pGEMHE–mCherry–Rab11a$^{S25N}$ and pCS2-EGFP-UtrCH have been previously described[19,20].

These constructs were linearized with AscI (NsiI for Utrophin). Capped mRNA was synthesized using T7 polymerase (mMessage mMachine kit, Ambion), and dissolved in 6–11 μl water. mRNA concentrations were determined using a NanoDrop spectrophotometer system (Thermo Scientific).

**Image analysis.** Zen software (Zeiss) was used to crop the centre of oocytes and Fiji/ImageJ was used to crop the cortex of the oocytes; Imaris software (Bitplane) was then used to analyse all images. The Spots detection function was used to detect and count central cortical granules in three dimensions by setting the detection parameters to 500 or 700 nm diameter. For total intensity measurements at the cell cortex, the Surfaces function was used to create a matrix covering the fluorescence at the plasma membrane, and the sum intensity was measured. For high spatial and temporal imaging, Rab27a vesicles were detected as 600-700 nm diameter spots and Rab11a vesicles were detected as 900 nm diameter spots and tracked. To determine hitchhiking Rab27a spots, an Imaris 'Spots Colocalize' MatLab XTension was used, and 1.5 μm was determined as the most accurate distance threshold for colocalization. The minimum track length for further analysis was set as five frames.

**Local mean squared displacement analysis.** We defined a set of squared displacements at each time point $t$ and lag $\tau$ for a given temporal analysis window of length $2\Delta t + 1$ as:

$$\Delta r^2(t,\tau) = \{t' \in [t - \Delta t, t + \Delta t], ||r(t' + \tau) - r(t')||^2\} \qquad (1)$$

As many of the switches between active and diffusive motion states along tracks were extremely transient ($<10$ s), the square displacements were computed using an analysis window of $2\Delta t + 1 = 9$ time points with maximum lag $\tau$ of 6 time points (illustrated in Supplementary Movie 10). The following second-order polynomial model with no constant term $a(t)\tau^2 + b(t)\tau$ was then robustly fitted using an iterative reweighted least square procedure to $\Delta r^2(t,\tau)$ at each time point $t$. We use the two parameters to discriminate between the types of motion each particle undergoes in its lifetime. To simplify, we refer to $a(t)$ as the 'velocity parameter' in μm$^2$ s$^{-2}$ and $b(t)$ as the 'diffusion parameter' in μm$^2$ s$^{-1}$.

To distinguish between active and diffusive motions, and between low and high diffusive motion (also termed 'anomalous diffusion'), a threshold on each parameter was set by calculating the 99.9 percentile of the distribution of control trajectories which did not display any active transport. We finally used these two thresholds to classify active (all steps with 'high speed') versus non-directed motion (all steps with 'low speed') and high diffusion versus low diffusion in the parameter space in the MSD scatter plots (Supplementary Figs 3e-h and 5b,c).

To test the reliability of our analysis, we repeated the MSD analysis but defined the two thresholds as the 99.9 percentile of the velocity and diffusion parameters of all Rab27a trajectories from control oocytes (active + non-active) using a robust estimation of the mean and variance (using the median and median of absolute deviation, respectively) assuming a Gaussian distribution (not shown). This produced a less stringent definition of active motions, but did not change the results of the experiment.

**Statistics and curve fitting.** Statistical testing was performed with InStat or Excel. Student's unpaired *t*-test was used to compare two normally distributed data sets, and Mann–Whitney's *U*-test for two non-parametric data sets. Kruskal–Wallis' ANOVA test was used to compare more than two data sets. Fischer's exact test was used to compare proportions of active steps. Tukey box plots show the median (line), mean (small square), interquartile range, and the 5th and 95th (whiskers). Origin Pro 8.0 was used for curve fitting. To fit central cortical granule curves, a Fast Fourier Transform filter was applied. To determine the start and end of the cortical granule decline period in the centre of the cell, the derivative of the smoothed curve was calculated. To fit the curves of cortical recruitment, Rab27a intensity increase at the cortex was fitted with a single exponential curve. Prior to the increase, at NEBD ± 30 min, the data were best fitted with a linear curve. MatLab and Octave were used to write the MSD analysis software. Significance levels: ns non-significant, *$P<0.05$, **$P<0.01$, ***$P<0.001$.

**Data availability.** The data that support the findings of this study are available within the article and its supplementary information files or from the corresponding author on reasonable request.

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

## Acknowledgements

We thank John Hammer III and Miguel Seabra for reagents, and the LMB light microscopy team and animal facilities for assistance. With thanks to members of the Schuh lab for helpful discussions. This study received financial support from the European Community's Seventh Framework Programme (FP7/2007-2013) under grant agreement no 241548 and from the Medical Research Council under grant MC_U105192711.

## Author contributions

L.P.C. performed all experiments and analysed the data. J.B. performed the MSD trajectory analysis with input from L.P.C. L.M.B. provided initial findings and a method for analysing movies. L.P.C. and M.S. designed the experiments and wrote the manuscript with input from all authors. M.S. supervised the study and acquired the images of actin with Rab27a.

**Additional information**

**Competing financial interests:** The authors declare no competing financial interests.

**Publisher's note**: 

