## [Peer Review File · Nature Communications]

Reviewer #1 (Remarks to the Author)

In their manuscript, Cheeseman and coworkers describe the mechanism by which cortical granules are transported to the cortex of mouse eggs, essential to prevent polyspermy. They show that Rab27a marks cortical granules and is required for their transport. They further show that this transport is actin-dependent and is in part mediated by myosin Va and to the other part by 'hitchhiking' on Rab11a vesicles.

These findings provide important new insights into an essential mechanism relevant to human infertility. The experiments are very well designed, the results shown are striking, and are described very clearly. Therefore, I highly recommend this manuscript for publication in Nature Communications. However, I would have a few comments, which should be addressed prior publication.

Major comment:

The authors are very strong about their 'hitchhiking' model that is even referred to in the title. However, I would like them to consider an alternative: the Schuh laboratory previously proposed a model for long-range transport of Rab11a vesicles. In this model Rab11a vesicles recruiting fmn2, spir1/2 and myosin Vb organize a dynamic actin network. In this context, the actin network organized around Rab11a nodes may be considered as the substrate on which Rab27a granules are transported by myosin Va. Thus, the 'hitchhiking' events may rather be pauses between bursts of rapid myosin Va-driven transport. Indeed, the scheme presented on Figure 6 is suggestive of such mechanism, but this is not discussed in the manuscript.

Could the authors discuss this alternative model? For example, would it be possible to estimate how much 'hitchhiking', i.e. co-localization between Rab11a and Rab27a would be expected when considering a short pause between myosin Va-driven transport phases? In this context, it would also be interesting to address whether Rab27a granules can only move on the actin network organized by Rab11a or can they also move on filaments independent of Rab11a?

Minor comments:

1. Table 1 shows values with extreme precision, disproportional to their spread (e.g. 28.25 +/- 33.01 min). This is unnecessary and misleading. Values should be rounded.
2. It would be important to show the number, not only the percentage, of granules quantified in Figure 1b.
3. For the MSD, the number of steps over which the values are calculated is unusually small. Could the authors test whether the time resolution of the data and the time window used is sufficient to support their conclusions?

Reviewer #2 (Remarks to the Author)

Cheeseman et al., report that mouse cortical granules are driven to the egg cortex by two mechanisms: 1) along actin filaments mediated by myosin Va; and 2) in association with Rab11a vesicles driven by myosin Vb. The manuscript is well-written, the studies are original and will be of considerable interest to the field. The experiments are thorough, controlled, well-executed and statistically sound. The authors provide a perspective on the field of cortical granule biology and draw appropriate conclusions that are thoughtfully discussed. A few points that the authors may wish to consider:

1. How might the Rab27a positive vesicles that are not cortical granules affect interpretation of results?

2. Fig. 2: Why doesn't Rab27a completely rescue translocation?

3. Fig. 6: Is the post-fertilization modification of the zona pellucida delayed or absent in Rab27aAsh/Ash mice?

Reviewer #3 (Remarks to the Author)

This is a very interesting and exciting study that is largely well supported by the data presented and should be of broad interest to the journal's readership. While the story is fairly complicated, the presentation is extremely clear and the figures are superb. The depth of some of the analyses is really very impressive. Overall I strongly support the publication of this study in NC. I do, however, have a few questions/comments that should be addressed prior to publication.

(1) Perhaps my only significant concern revolves around the specificity of the myosin Va and Vb dominant negative tail constructs used by the authors to parse out the relative contributions of these two motors to the transport of the cortical granules. Historically, the cargo binding tail domains of organelle motors has been used as dominant negative constructs because they contain the cargo-specific binding sites (so that, when over-expressed, they displace the endogenous full length motor from its cargo). The use of such constructs to disrupt the functions of type V myosins has, however, become more complicated because of the realization that these myosins are regulated by intramolecular folding driven by the tail domain. Specifically, in the absence of cargo these molecules fold into a compact, inactive conformation that is stabilized by specific interactions between the tail domains and the head domains (work from the Sellers, Trybus and Ikebe labs). Cargo either induces the unfolding of myosin V into its open, active conformation or kinetically traps the open conformation. The possible problem as regards this study is that tail domains can inhibit head domains in trans, and, most problematic for this study, in an isoform-independent way. For example, Ehlers and colleagues have shown that the tail domain of myosin Vb can inhibit intact myosin Va to some extent. I think the authors may be able to argue away this issue with existing/prior data. If not, they should perform an experiment that mitigates this issue (e.g. over express versions of tail domains containing known point mutations that abrogate interaction with the head but not cargo; use RNAi, which should be isoform specific). In a related issue, myosin Va has been reported to interact with many Rab GTPases, including Rab11a, which previously was considered to interact only with myosin Vb. Also, the over-expression of the specific myosin Va tail domains used here (which include most of the central coiled coil domain) could in principal soak up dynein light chain 8, which could have consequences for other pathways. I am not that concerned with these two latter issues but perhaps a disclaimer could be included.

(2) When referencing the studies showing the Rab27a-dependent recruitment of myosin Va to melanosomes, the authors should cite the following four papers, which were published almost simultaneously in 2002: (1) Fukuda et al. *J. Biol. Chem.* 277: 12432-12436, (2) Wu et al *Nat. Cell Biol.* 4: 271-278, (3) Nagashima et al *FEBS Letts* 517: 233-238, and (4) Strom et al *J. Biol. Chem.* 277: 25423-25430.

(3) In some cases (e.g. melanosomes), the interaction of the Rab GTPase with myosin Va requires a specific Rab effector to bridge their indirect interaction (in that case, melanophilin). In other cases (e.g. myosin Vb, Rab11a, and Rab FIPs), it may or may not, while in the case of certain myosin V: Rab interactions in yeast the interaction is clearly direct. Although it is outside the scope of this work to resolve this question in mouse eggs, it might be nice to discuss this issue in a sentence or two, especially as it raises the possibility that another mouse mutant (one in a Rab effector) would also exhibit a severe defect in the block to multiple sperm entry.

(4) To my knowledge, litter sizes and pup health are essentially normal for ashen mice. Does this fit with their last piece of data on the pronounced defect in ashen mice eggs?

(5) The authors need to be much more careful with the use of the term "docking", which means the physical tethering of the vesicle to the membrane that precedes priming and SNARE-dependent fusion. They never look in a way (basically EM or TIRF) that can prove docking (take a look at Gilliam Griffith's and Pierre Henkart's papers on Rab27 in T cells in *JCB* 2002 and subsequent papers from the Izumi lab on Rab27a in Beta cells).

(6) On page 7, lines 2-4, for the sake of transparency, the sentence should make it clear that

while rescue resulted in significant improvements relative to ashen, it did not restore parameters to control levels. Would GTP-locked Rab27a do the trick?

(7) On page 8, line 24 they refer to movement of granules on actin filaments but the structure in the image (3G) must be an actin filament bundle I think.

(8) On page 10, lines 1-4 the authors should state a little more clearly the results in 4E and 4F ("similar values" is too vague). It looks like cortical granules associated with both Rabs move at a speed that is intermediate between the speed of granules associated with only Rab27a or with only Rab11a, right?

Re: Manuscript NCOMMS-16-13772A

“A hitchhiking and a non-hitchhiking pathway translocate cortical granules to prevent polyspermy in mouse oocytes”

Liam P. Cheeseman, Jérôme Boulanger, Lisa M. Bond and Melina Schuh

Reviewer #1: Expert in actin, oocyte, mitosis

(Remarks to the Author):

In their manuscript, Cheeseman and coworkers describe the mechanism by which cortical granules are transported to the cortex of mouse eggs, essential to prevent polyspermy. They show that Rab27a marks cortical granules and is required for their transport. They further show that this transport is actin-dependent and is in part mediated by myosin Va and to the other part by 'hitchhiking' on Rab11a vesicles.

These findings provide important new insights into an essential mechanism relevant to human infertility. The experiments are very well designed, the results shown are striking, and are described very clearly. Therefore, I highly recommend this manuscript for publication in Nature Communications. However, I would have a few comments, which should be addressed prior publication.

> We thank the reviewer for his/her time and kind comments. We are very pleased that the reviewer values the quality of our results and that he/she finds that this work provides important new insights in the context of human infertility.

Major comment:

The authors are very strong about their 'hitchhiking' model that is even referred to in the title. However, I would like them to consider an alternative: the Schuh laboratory previously proposed a model for long-range transport of Rab11a vesicles. In this model Rab11a vesicles recruiting fmn2, spir1/2 and myosin Vb organize a dynamic actin network. In this context, the actin network organized around Rab11a nodes may be considered as the substrate on which Rab27a granules are transported by myosin Va. Thus, the 'hitchhiking' events may rather be pauses between bursts of rapid myosin Va-driven transport. Indeed, the scheme presented on Figure 6 is suggestive of such mechanism, but this is not discussed in the manuscript.

Could the authors discuss this alternative model? For example, would it be possible to estimate how much 'hitchhiking', i.e. co-localization between Rab11a

and Rab27a would be expected when considering a short pause between myosin Va-driven transport phases? In this context, it would also be interesting to address whether Rab27a granules can only move on the actin network organized by Rab11a or can they also move on filaments independent of Rab11a?

> This is an interesting suggestion, and we think it is very close to the model we describe. Whether the association between Rab11a and Rab27a happens ‘by accident’ between bursts of myosin Va-dependent movement is hard to determine. However when Rab11a is inhibited (Fig 4 h, i) or when myosin Vb is inhibited (new Supplementary Fig S6 a, b, c), the translocation of cortical granules is impaired, which suggests that the hitchhiking actively contributes to cortical granule translocation rather than merely being a by-product. We also find that myosin Va-dependent movement alone is not sufficient for efficient translocation of cortical granules (Fig 4 h, i; Fig 5 f, g, h, i, Supplementary Fig S6 a, b, c).

To bolster our model, we have now included a new video (Supplemental Movie 9) where we show the behaviour of cortical granules when myosin Vb is inhibited. One can observe that although movements of Rab27a are very limited due to a static actin network, there are still noticeable bursts of short, rapid movement, which are a hallmark of myosin Va activity. This suggests that cortical granules are able to move along microfilaments independently of Rab11a, powered by myosin Va. Accordingly, we have added a sentence in the Results, page 13, line 18-23:

“These fast motions still took place when myosin Vb was inhibited, suggesting that a different motor powered their movement separately from myosin Vb and Rab11a (Supplementary Movie S9). We therefore queried whether this myosin Vb- and Rab11a-independent pathway might be powered by myosin Va.”

We have also added further discussion relative to the alternative model in the text, p15, line 19-24:

“An alternative interpretation is that the hitchhiking is merely a by-product of random encounters between Rab11a and Rab27a. However, the inhibition of Rab11a or myosin Vb both prevented efficient granule translocation (Fig. 4h,i; Supplementary Fig. S6), indicating that Rab11a hitchhiking actively contributes to the transport of cortical granules and is a bona fide pathway.”

Minor comments:

1. Table 1 shows values with extreme precision, disproportional to their spread (e.g. 28.25 +/- 33.01 min). This is unnecessary and misleading. Values should be rounded.

> We agree, and have rounded the values in the table accordingly.

2. It would be important to show the number, not only the percentage, of granules quantified in Figure 1b.

> This is a very good suggestion, and we have now added this information to Figure 1b.

3. For the MSD, the number of steps over which the values are calculated is unusually small. Could the authors test whether the time resolution of the data and the time window used is sufficient to support their conclusions?

> This is an important point. It has previously been shown that while a short window of MSD analysis is not optimal for estimating the MSD parameters, it can be used effectively to detect changes in trajectory motion (Michalet, 2010. Phys Rev E Stat Nonlin Soft Matter Phys. Oct;82(4 Pt 1):041914).

For this experiment, the acquisition rate (every 2-2.5 sec) could not be lowered without compromising signal to noise ratio. In the context of the local MSD analysis, one must make a compromise in terms of the length of the window of analysis. A shorter window increases the variance of the data, with MSD data fluctuating considerably (the variation can be seen in the spread of the parameters in MSD scatter plots). A longer time window causes an overall smoothing of the data, meaning it becomes more difficult to identify instances of switching between directed movement and diffusion. To this end, we tested multiple lengths for the MSD analysis window to find a suitable compromise visually reflecting the behaviour of the particles, which we illustrate in Supplementary Video S10.

Reviewer #2 : Expert in gamete structure/recognition

(Remarks to the Author):

Cheeseman et al., report that mouse cortical granules are driven to the egg cortex by two mechanisms: 1) along actin filaments mediated by myosin Va; and 2) in association with Rab11a vesicles driven by myosin Vb. The manuscript is well-written, the studies are original and will be of considerable interest to the field. The experiments are thorough, controlled, well-executed and statistically sound. The authors provide a perspective on the field of cortical granule biology and draw appropriate conclusions that are thoughtfully discussed. A few points that the authors may wish to consider:

> We thank the reviewer for his/her time and feedback on the manuscript,

and were very pleased that he/she only had minor comments.

1. How might the Rab27a positive vesicles that are not cortical granules affect interpretation of results?

> We report in Figure 1b that ~25% of Rab27a puncta do not colocalise with lens culinaris agglutinin. Whilst this is a significant proportion of Rab27a puncta, we have several reasons to conclude that these do not significantly affect the outcome of the results.

First, although LCA is the only known marker with specificity for cortical granules, it is still unclear exactly how specific the reporter is. It binds a mannose motif that may not be present on all cortical granules, which could account for discrepancies between Rab27a localisation and LCA staining. This could further be influenced by the degree of maturation of the granules, which may be positive for Rab27a but not LCA.

Second, we use enrichment at the cortex and a reduction in number in the oocyte centre as functional readouts in our experiments as these are hallmarks of cortical granule behaviour. We do not observe a subpopulation of Rab27a granules which defies this behaviour, indicating that these Rab27a-positive granules still behave like cortical granules despite being LCA-negative.

We have now added a comment concerning this discrepancy in the text (p. 5, line 5-9):

“Interestingly, although 25% of Rab27a puncta did not stain positive for LCA, we did not observe a subpopulation that remained in the oocyte centre, suggesting that the LCA-negative granules also displayed the hallmarks of cortical granule behaviour. It is possible that the variability in staining may reflect different stages of granule maturation.”

2. Fig. 2: Why doesn't Rab27a completely rescue translocation?

> Rab27a^{Ash/Ash} transcripts are still expressed in Ashen tissue (Wilson et al., 2000), suggesting the truncated version of Rab27a is still expressed. It is possible that this inactive form acts as a dominant negative and prevents a full rescue of the Ashen phenotype.

We have attempted to achieve a fuller rescue by expressing a constitutively active variant of Rab27a, however this construct failed to localise to cortical granules, suggesting it is not active in oocytes.

In our experiments, we expressed exogenous Rab27a for 4-5 hours before releasing the Ashen oocytes from prophase arrest. As with the majority of rescue experiments, it is possible that this is an insufficient expression to fully rescue the Ashen phenotype. However, prolonged prophase arrest causes defects in oocytes, so we did not express Rab27a for longer periods.

3. Fig. 6: Is the post-fertilization modification of the zona pellucida delayed or absent in Rab27aAsh/Ash mice?

> This is an excellent question. We have now quantified cortical granules remaining in the oocyte centre following fertilisation in these cells, and included the data in Figure 6d (see below). We find that after fertilisation, cortical granule numbers are much higher in mutant oocytes compared to the controls, indicating that the cortical reaction must be incomplete or even absent, consistent with our observation that a higher number of sperm penetrate the zona in Ashen mutant oocytes. We have alternatively considered measuring the intensity of cortical granules at the cortex in control and Ashen oocytes following fertilisation. However, the interpretation of the results would have been difficult because cortical granules fail to enrich at the cortex in Ashen oocytes.

Reviewer #3 ; Expert in myosin V

(Remarks to the Author):

This is a very interesting and exciting study that is largely well supported by the data presented and should be of broad interest to the journal's readership. While the story is fairly complicated, the presentation is extremely clear and the figures are superb. The depth of some of the analyses is really very impressive. Overall I strongly support the publication of this study in NC. I do, however, have a few questions/comments that should be addressed prior to publication.

> We thank the reviewer for taking the time to read our manuscript, and for the very encouraging comments. We are pleased that we could convey our results in a clear manner.

(1) Perhaps my only significant concern revolves around the specificity of the myosin Va and Vb dominant negative tail constructs used by the authors to parse out the relative contributions of these two motors to the transport of the cortical granules. Historically, the cargo binding tail domains of organelle motors has been used as dominant negative constructs because they contain the cargo-specific binding sites (so that, when over-expressed, they displace the endogenous full length motor from its cargo). The use of such constructs to disrupt the functions of type V myosins has, however, become more complicated because of the realization that these myosins are regulated by intramolecular folding driven by the tail domain. Specifically, in the absence of cargo these molecules fold into a compact, inactive conformation that is stabilized by specific interactions between the tail domains and the head domains (work from the Sellers, Trybus and Ikebe labs). Cargo either induces the unfolding of myosin V into its open, active conformation or kinetically traps the open conformation. The possible problem as regards this study is that tail domains can inhibit head domains in trans, and, most problematic for this study, in an isoform-independent way. For example, Ehlers and colleagues have shown that the tail domain of myosin Vb can inhibit intact myosin Va to some extent. I think the authors may be able to argue away this issue with existing/prior data. If not, they should perform an experiment that mitigates this issue (e.g. over express versions of tail domains containing known point mutations that abrogate interaction with the head but not cargo; use RNAi, which should be isoform specific).

> We thank the reviewer for bringing this important matter to our attention. To address concerns regarding the cross-inhibition of the dominant negative constructs used in our experiments, we have now included data for the translocation of cortical granules in oocytes depleted for myosin Va and myosin Vb by RNAi, which we include in Supplementary Figure S6 (see below). Both the depletion of myosin Va and of myosin Vb significantly

inhibited the characteristic decrease in granule numbers in the cell centre, indicating that both myosins are indeed involved in the translocation process. The quantification at the cortex also suggested an inhibition of granule recruitment, however the large variability of LCA staining between cells means that it was not statistically significant. We have added a section in the Results to describe this (p. 14, line 1-12):

“To test if both myosin Va and myosin Vb were involved in cortical granule translocation, we depleted each protein using siRNA in oocytes cultured from follicle cells. The cells were then fixed either before NEBD, or after being released from prophase arrest for 5 hours, and then stained with LCA (Supplementary Fig. S6a). We found that depletion of either myosin caused the number of remaining cortical granules in the cell centre to increase significantly compared to controls (Supplementary Fig. S6b). The intensity at the cortex was also lower than in controls when either myosin was depleted (Supplementary Fig. S6c), although significance was not attained due to high staining variability between cells. These results suggest that both myosin Va and myosin Vb are involved in the translocation of cortical granules.”

In a related issue, myosin Va has been reported to interact with many Rab GTPases, including Rab11a, which previously was considered to interact only with myosin Vb. Also, the over-expression of the specific myosin Va tail domains used here (which include most of the central coiled coil domain) could in principal soak up dynein light chain 8, which could have consequences for other pathways. I am not that concerned with these two latter issues but perhaps a disclaimer could be included.

> We thank the reviewer for highlighting these points. We have now included disclaimers concerning these two points in the text.

Regarding the inhibition of myosin Va potentially affecting Rab11a, we have shown in a previous study that the Myosin Va tail domain did not affect Rab11a motility or its associated functions in oocytes, and have added this to the text.

- P12, line 23: “[...] which did not affect Rab11a motility in a previous study (Holubcova et al 2013).”

Regarding the potential inhibition of pathways involving dynein light chain 8, we show that microtubules are not involved in the translocation process (Fig. 3a, b), suggesting that dynein LC8 inhibition is unlikely to affect granule translocation.

- P12-13: “We note that the MyoVa^{LT} construct used here is reported to bind dynein light chain 8^{30,31}. However, given that our results demonstrated clearly that the translocation of cortical granules is independent of microtubules (Fig. 3a,b), this potential secondary effect of MyoVa^{LT} is unlikely to have an influence on cortical granule behaviour.”

(2) When referencing the studies showing the Rab27a-dependent recruitment of myosin Va to melanosomes, the authors should cite the following four papers, which were published almost simultaneously in 2002: (1) Fukuda et al. J. Biol. Chem. 277: 12432-12436, (2) Wu et al Nat. Cell Biol. 4:271-278, (3) Nagashima et al FEB Letts 517: 233-238, and (4) Strom et al J. Biol. Chem. 277:25423-25430.

> We thank the reviewer for raising this point. We have now added these citations to the main text (p. 18, line 1).

(3) In some cases (e.g. melanosomes), the interaction of the Rab GTPase with myosin Va requires a specific Rab effector to bridge their indirect interaction (in that case, melanophilin). In other cases (e.g. myosin Vb, Rab11a, and Rab FIPs), it may or may not, while in the case of certain myosin V: Rab interactions in yeast the interaction is clearly direct. Although it is outside the scope of this work to resolve this question in mouse eggs, it might be nice to discuss this issue in a sentence or two, especially as it raises the possibility that another mouse mutant (one in a Rab effector) would also exhibit a severe defect in the block to multiple sperm entry.

> This is a good point which will be of importance in future studies. We now discuss this topic in the Discussion, p18, line 3-9:

“Since Rab27a and myosin Va are not thought to bind directly^{27,43,49,50}, it is likely that an adaptor links the two proteins, similarly to the tripartite

complexes reported in melanocytes and cytotoxic T lymphocytes^{27,45,48-50,52,54}. The identification of the adaptor protein will be of importance in future studies, and raises the interesting possibility that a mouse mutant for this adaptor would display severe defects in the efficiency of the zona block to sperm entry.”

(4) To my knowledge, litter sizes and pup health are essentially normal for ashen mice. Does this fit with their last piece of data on the pronounced defect in ashen mice eggs?

> Indeed, pup health is normal in Ashen mice. Consistent with our data, a previous study reported that litters from two different Ashen mouse strains have lower litter sizes compared to their wild-type background strains (discussed on page 19, lines 1-2: “A previous study reported a 20 to 50% reduction in litter size in Ashen female mice compared to their background strain (Tolmachova et al. 2004).”). We were not able to investigate this point ourselves, as we only bred $Rab27a^{+/Ash}$ and $Rab27a^{Ash/Ash}$ mice, with the heterozygote used as control.

A further point to consider is that other blocks to polyspermy are still likely to occur in Ashen mice, which may explain the mild phenotype. This includes the limitation of the number of sperm that travel through the reproductive tract and ultimately reach the eggs, and the block at the plasma membrane, which acts after the cortical reaction.

(5) The authors need to be much more careful with the use of the term "docking", which means the physical tethering of the vesicle to the membrane that precedes priming and SNARE-dependent fusion. They never look in a way (basically EM or TIRF) that can prove docking (take a look at Gilliam Griffith's and Pierre Henkart's papers on Rab27 in T cells in JCB 2002 and subsequent papers from the Izumi lab on Rab27a in Beta cells).

> We thank the reviewer for raising this issue. We have now replaced the instances of “docking” with more cautious terms (usually replaced by “association with the plasma membrane” or “maintenance at the cortex”).

(6) On page 7, lines 2-4, for the sake of transparency, the sentence should make it clear that while rescue resulted in significant improvements relative to ashen, it did not restore parameters to control levels. Would GTP-locked Rab27a do the trick?

> We thank the reviewer for his/her suggestion. We attempted to achieve a more complete rescue of translocation in Ashen oocytes by expressing a constitutively active variant of Rab27a ($Rab27a^{Q78L}$). However, the construct did not localise to cortical granules, suggesting that it is inactive in

oocytes. As with all rescue experiments, it is possible that by allowing for more expression of the functional Rab27a (we expressed for 4-5 hours before prophase release), we could have achieved a more complete rescue. However, prolonged prophase arrest using dbcAMP also tends to produce defects in mouse oocytes, so we did not express for longer.

In addition, Rab27a^{Ash/Ash} transcripts are still expressed in Ashen tissue (Wilson et al., 2000), suggesting the truncated version of Rab27a is still expressed. It is possible that this inactive form acts as a dominant negative and prevents a full rescue of the Ashen phenotype.

We have now altered the description of our results to make it clear the rescue was only partial; p. 7, line 9: “This significantly rescued the translocation of cortical granules to the cortex and reduced the number of granules in the cell centre, although these did not reach control levels.”

(7) On page 8, line 24 they refer to movement of granules on actin filaments but the structure in the image (3G) must be an actin filament bundle I think.

> We have now altered the wording for this, and replaced “actin filament” with “actin structure” (p. 9, line 5). Similarly, we have altered the legend of Figure 3H to reflect this.

(8) On page 10, lines 1-4 the authors should state a little more clearly the results in 4E and 4F ("similar values" is too vague). It looks like cortical granules associated with both Rabs move at a speed that is intermediate between the speed of granules associated with only Rab27a or with only Rab11a, right?

> We have now altered the wording to produce a more precise description of the data. P. 10, line 8-9: “[...] were of intermediate value between those of Rab27a alone and of Rab11a vesicles themselves (Fig. 4e,f)”.

Reviewer #1 (Remarks to the Author)

The revised version of the manuscript and the rebuttal letter addresses all of my concerns. Thus, I now fully support the publication of the manuscript in Nature Communications.

Reviewer #2 (Remarks to the Author)

I have no further comments to offer.

Reviewer #3 (Remarks to the Author)

Great paper and great revisions. Congrats.